# INTENDD: A Unified Contrastive Learning Approach for Intent Detection and Discovery

**Bhavuk Singhal**
Uniphore Inc.
bhavuk.singhal@uniphore.com

**Ashim Gupta**
University of Utah
ashim@cs.utah.edu

**Shivasankaran V P**
Stony Brook University
svanajapandi@cs.stonybrook.edu

**Amrith Krishna***
amrith.tech
ask@amrith.tech

## Abstract

Identifying intents from dialogue utterances forms an integral component of task-oriented dialogue systems. Intent-related tasks are typically formulated either as a classification task, where the utterances are classified into predefined categories or as a clustering task when new and previously unknown intent categories need to be discovered from these utterances. Further, the intent classification may be modeled in a multiclass (MC) or multilabel (ML) setup. While typically these tasks are modeled as separate tasks, we propose INTENDD a unified approach leveraging a shared utterance encoding backbone. INTENDD uses an entirely unsupervised contrastive learning strategy for representation learning, where pseudo-labels for the unlabeled utterances are generated based on their lexical features. Additionally, we introduce a two-step post-processing setup for the classification tasks using modified adsorption. Here, first, the residuals in the training data are propagated followed by smoothing the labels both modeled in a transductive setting. Through extensive evaluations on various benchmark datasets, we find that our approach consistently outperforms competitive baselines across all three tasks. On average, INTENDD reports percentage improvements of 2.32 %, 1.26 %, and 1.52 % in their respective metrics for few-shot MC, few-shot ML, and the intent discovery tasks respectively.

## 1 Introduction

Intents form a core natural language understanding component in task-oriented dialogue (ToD) systems. Intent detection and discovery not only have immense utility but are also challenging due to numerous factors. Intent classes vary vastly from one use case to another, and often arise out of business needs specific to a particular product or organization. Further, modeling requirements might necessitate considering fine-grained

and semantically-similar concepts as separate intents (Zhang et al., 2021c). Overall, intent-related tasks typically are expected to be scalable and resource efficient, to quickly bootstrap to new tasks and domains; lightweight and modular for maintainability across domains and expressive to handle large, related often overlapping intent scenarios (Vulić et al., 2022; Zhang et al., 2021a).

INTENDD proposes a unified framework for intent detection and discovery from dialogue utterances from ToD systems. The framework enables the modeling of various intent-related tasks such as intent classification, both multiclass and multilabel, as well as intent discovery, both unsupervised and semi-supervised. In Intent detection (classification), we expect every class to have a few labeled instances, say 5 or 10. However, in intent discovery, not all classes are expected to have labeled instances and may even be completely unlabeled.

Recently, intent-related models focus more on contrastive representation learning, owing to the limited availability of labeled data and the presence of semantically similar and fine-grained label space(Kumar et al., 2022; Zhang et al., 2021c). Similarly, a common utterance encoder forms the backbone of INTENDD, irrespective of the task. The utterance encoder is learned by updating the parameters of a general-purpose pre-trained encoder using a two-step contrastive representation learning process. First, we adapt a general-purpose pre-trained encoder by using unlabelled information from various publicly available intent datasets. Second, we update the parameters of the encoder using utterances from the target dataset, on which the task needs to be performed, making the encoder specialize on the corpus. Here, we use both labeled and unlabelled utterances from the target dataset, where pseudo labels are assigned to the latter.

For intent classification, both multiclass and multilabel, INTENDD consists of a three-step pipeline. It includes training a classifier that uses the rep-

---

*Work done while at Uniphore

resentation from the encoder as its feature representation, followed by two post-processing steps in a transductive setting. Specifically, a multilayer perceptron-based classifier is trained by stacking it on top of the utterance representation from our encoder. The post-processing steps consider the target corpus as a graph in a transductive setting. The first postprocessing step involves propagating the residual errors in the training data to the neighbors. The second one further performs label smoothing by propagating the labels obtained from the previous step. Both these steps are performed using Modified Adsorption, an iterative algorithm that enables controlling the propagation of information that passes through a node more tightly (Talukdar and Pereira, 2010).

**Major contributions:** INTENDD reports performance improvements compared to that of competitive baselines in all the tasks and settings we experimented with, including multiclass and multilabel classification in few-shot and high data settings; unsupervised and semi-supervised intent discovery. Our two-step post-processing setup for intent classification leads to statistically significant performance improvements to our base model. While existing intent models focus primarily on better representation learning and data augmentation, we show that classical transductive learning approaches can help improve the performance of intent models even in fully supervised settings. Finally, we show that with a careful construction of a graph structure in a transductive learning setting in terms of both edge formation and edge weight formation can further improve our outcomes.

## 2 INTENDD

INTENDD consists of a two-step representation learning module, a classification module, and an intent detection module. We elaborate on each of these modules in this section.

### 2.1 Continued Pretraining

We start with a general-purpose pre-trained model and use it as a cross-encoder for the continued pretraining (Gururangan et al., 2020). We start with a standard general-purpose pre-trained model as the encoder. We follow Zhang et al. (2021c) for our pretraining phase where the model parameters are updated both using a combination of token-level masked language modeling loss and a sentence-level self-supervised contrastive loss. For a batch

of $K$ sentences, we compute the contrastive loss (Wu et al., 2020; Liu et al., 2021) as follows

$$\mathcal{L}_{sscl} = -\frac{1}{K} \sum_{i=1}^{K} log \frac{exp(sim(h_i, \bar{h}_i)/\tau)}{\sum_{j=1}^{K} exp(sim(h_i, \bar{h}_j)/\tau)} \tag{1}$$

For a sentence $x_i$, we obtain a masked version of the sentence $\bar{x}_i$, where a few tokens of $x_i$ are randomly masked. Further, we dynamically mask tokens such that each sentence has different masked positions across different training epochs. In $\mathcal{L}_{sscl}$, $h_i$ is the representation of the sentence $x_i$ and $\bar{h}_i$ is the representation of the $\bar{x}_i$. $\tau$ is the temperature parameter that controls the penalty to negative samples and $sim(.,.)$ denotes the cosine similarity between two vectors. The final loss $\mathcal{L}_{pretraining}$ is computed as $\mathcal{L}_{pretraining} = \mathcal{L}_{sscl} + \lambda \mathcal{L}_{mlm}$. Here, $\mathcal{L}_{mlm}$ is token level masked language modelling loss and $\lambda$ is a weight hyper-parameter.

### 2.2 Corpus-specialized Representation Learning

The pretraining step uses unlabelled sentences from publicly available intent datasets which should ideally expose a pre-trained language model with utterances in the domain. Now, we consider contrastive representation learning using the target dataset on which the task needs to be performed.

Consider a dataset $\mathcal{D}$ with a total of $N$ unlabelled input utterances. Here, assuming $\mathcal{D}$ to be completely unlabeled, we first assign pseudo labels to each of the utterances in $\mathcal{D}$. Using the pseudo labels, we learn corpus-level contrastive representation by using supervised contrastive loss (Khosla et al., 2020). The pseudo labels are assigned by first finding clusters of utterances by using a community detection algorithm, 'Louvain' (Blondel et al., 2008). Community detection assumes the construction of a graph structure. We form a connected weighted directed graph $G_{\mathcal{D}}(V_{\mathcal{D}}, E, W)$, the input utterances in $\mathcal{D}$ form the nodes in $G_{\mathcal{D}}$. We identify lexical features in the form of word-level n-grams.

We identify keyphrases that are representative of the target corpus on which the representation learning is performed. The keyphrases are obtained by finding word-level n-grams that have a high association with the target corpus, as compared to the likelihood of finding those in other arbitrary corpora. Here, we obtain the pointwise mutual information (PMI) of the n-grams in the target corpus, based on the likelihood of the n-gram occurring in

the corpus, compared to a set of utterances formed via the union of the sentences in the target corpus and that in the corpora used during pretraining setup. Let $\mathcal{P}$ be the union of all the sentences in the corpora used in the pretraining step. Now, the PMI is calculated as

$$PMI(kp, \mathcal{D}) = \log df(kp, \mathcal{P} \cup \mathcal{D}) \times$$
$$\log \frac{df(kp, \mathcal{D})|\mathcal{P} \cup \mathcal{D}|}{df(kp, \mathcal{P} \cup \mathcal{D})|\mathcal{D}|} \quad (2)$$

Here, $df(kp, \mathcal{D})$ is the count of utterances in $\mathcal{D}$ that contain the keyphrase $kp$. $df(kp, |\mathcal{P} \cup \mathcal{D}|)$ is the frequency of the keyphrase from the combined collection $\mathcal{D}$ and $\mathcal{P}$. Here, we only consider those keyphrases which is present at least five times in $\mathcal{D}$. Moreover, the log frequency of the count of the keyphrase is also multiplied with PMI to avoid high scores for rare words (Jin et al., 2022). Further, the PMI value is multiplied by the square of the number of the words in the ngram so as to have higher scores for ngrams with larger values of $n$ (Banerjee and Pedersen, 2002). We validated this decision during preliminary experiments where we found that multiplying PMI with the square of the number of words generally worked better for the datasets considered in this work. That said, it's important to note that this design choice may vary in its necessity when applied to a different dataset, and its requirement should be established through empirical investigation.

Now, the keyphrases are used to construct $G_{\mathcal{D}}$. Two nodes have edges between them if they both have at least one common keyphrase. The edge weights are the sum of the keyphrase scores common between two nodes. The weight matrix $W$ is a $N \times N$ matrix representing the edge weights in the graph. $W$ is row-normalized using min-max normalization, a form of feature scaling. The graph $G_{\mathcal{D}}$ is then used to perform community detection using Louvain, a modularity-based community detection algorithm. Community membership is used to form clusters of inputs. Here, all the nodes in $G_{\mathcal{D}}$ that belong to the same cluster are assigned with a common (pseudo)-label.

**Louvain Method:** is a modularity-based graph partitioning approach for detecting hierarchical community structure (Blondel et al., 2008). Here, each utterance is considered a node in a graph and the edge weights capture the strength of the relation between node pairs. Louvain Method attempts to iteratively maximize the quality function it optimizes, generally modularity. While the approach may be started with any arbitrary partitioning of the graph, we start with each data point belonging to its own community (singleton communities). It then works iteratively in two phases. In the first phase, the algorithm tries to assign the nodes to their neighbors' community as long as that reassignment leads to a gain in the modularity value. The second phase then aggregates the nodes within a community and forms a super node, thus creating a new graph where each community in the first phase becomes a node in the second phase. The process iteratively continues until the modularity value can no longer be improved.

Until now, we were assuming $G_{\mathcal{D}}$ to be completely unlabeled. However, we are yet to discuss two crucial questions. One, how to incorporate labeled information for an available subset of utterances in a semi-supervised setup. Here, we need to ensure that nodes belonging to the same true label should not get partitioned into separate clusters. We merge those inputs with the same true label as a single node before constructing $G_{\mathcal{D}}$, and initialize Louvain with the graph structure so obtained. The merging of the utterances with a common label into a single node trivially ensures that no two utterances of the same label get partitioned into different clusters. Hence, we ensure that no two nodes with the same true label are assigned with different pseudo labels. However, at this stage, the pseudo-labels are obtained purely for representation learning. It is not intended to be representative of the real intent classes but is rather simply a partition based on the keyphrases in the utterances. Finally, Using the pseudo labels obtained via Louvain, we learn corpus-level contrastive representation by using supervised contrastive loss (Khosla et al., 2020). Here, during the representation learning each utterance is treated separately and we do not consider the merging that we performed for the community detection.

**Keyphrase selection for constructing $G_{\mathcal{D}}$:** While we have a list of n-grams, along with their feature scores. Here, we employ recursive feature elimination (RFE), a greedy feature elimination approach as our feature selection strategy. In RFE we start with a large set of features and greedily eliminate features, one at a time. We start with the top k features and perform the community detection using Louvain. We then start with the least

promising feature from our selected features and check if removing the feature leads to an increase in the overall modularity of the graph, as compared to the modularity when the feature was included. Here, the number of nodes in $G_{\mathcal{D}}$ remain the same, though the number of edges and their edge weights are dependent on the features. A single run of the Louvain algorithm has a complexity of O(n.logn), where n is the number of nodes. So in worst case, the time complexity for graph construction is $O(n.d.logn)$, where $d$ is the number of features. We perform the feature selection for a few fixed iterations. We incorporate some additional constraints to keep track of for the feature selection, which are as follows: The graph needs to remain a single connected graph and if the removal of a feature violates it, then we keep the feature. Second, in all the tasks we consider, we assume the knowledge of the total number of intents. Hence a feature, whose presence, even if contributes positively to modularity but results in increasing the gap between the total number of true intent classes and the number of clusters Louvain provides with it as the feature, then the feature is removed as well.

## 2.3 Intent Discovery

We perform intent discovery in both unsupervised and semi-supervised setups. Intent discovery is performed via clustering. Here, we start with the same graph construction as was used for Louvain in §2.2. The weight matrix $\mathbf{W}$ is row-normalized. Additionally, we obtain a similarity matrix $\mathbf{A}$ based on the cosine similarity between the utterance level encodings of two nodes. The encodings are obtained from the encoder learned in §2.2. We obtain a weighted average of the edge weights in $\mathbf{W}$ and $\mathbf{A}$. Specifically, the weights for the average is obtained via grid search and selects the configuration that optimizes the silhouette score, an intrinsic measure for clustering quality. The new graph will be referred to as $\mathcal{G}_{pred}$. With $\mathcal{G}_{pred}$, we perform Louvain again for intent discovery. The labeled nodes in a semi-supervised setup would be merged as a single node before running Louvain. When a new set of utterances arrive, these utterances are added as nodes in $\mathcal{G}_{pred}$. Their corresponding values in $\mathbf{A}$ are obtained based on their representation obtained from our encoder (§2.2). The corresponding values in $\mathbf{W}$ are obtained based on the existing set of ngrams and no new feature selection is performed.

## 2.4 Intent Classification

Irrespective of whether multiclass or multilabel setup, our base classifier is a multilayer perceptron comprising of a single hidden layer with nonlinearity. It uses the utterance level representation, learned in §2.1 and §2.2, as its input feature, which remains frozen during the training of the classifier. The classifier is trained using cross-entropy loss with label smoothing (Vulić et al., 2022; Zhang et al., 2021c). The activation function at the output layer is set to softmax and sigmoid for multiclass and multilabel classification respectively.

**Modified Adsorption (MAD)** is a graph-based semi-supervised transductive learning approach (Talukdar and Crammer, 2009). MAD is a variant of the label propagation approach. While label propagation (Zhu et al., 2003) forces the unlabeled instances to agree with their neighboring labeled instances, MAD enables prediction on labeled instances to vary and incorporates node uncertainty (Yang et al., 2016). It is expressed as an unconstrained optimization problem and solved using an iterative algorithm that guarantees convergence to a local optima (Talukdar and Pereira, 2010; Sun et al., 2016). The graph typically contains a few labeled nodes, referred to as seed nodes, and a large set of unlabelled nodes. The graph structure can be explicitly designed in MAD. The unlabelled nodes are typically assigned a dummy label. In MAD, a node actually is assigned a label distribution than a hard assignment of a label.

From a random walk perspective, it can be seen as a controlled random walk with three possible actions, each with predefined probabilities, all adding to one (Kirchhoff and Alexandrescu, 2011). The three actions involve a) continuing a random walk to the neighbors of a node based on the transition matrix probability, b) stopping and returning the label distribution for the node, and c) abandoning and returning an all-zero distribution or a high probability to the dummy label. Each of these components forms part of the MAD objective in the form of seed label loss, smoothness loss across edges, and the label prior loss. The objective is:

$$\arg\min_{\hat{\mathbf{Y}}} \sum_{l=1}^{K+1} \left[ \left\| \mathbf{S}\hat{\mathbf{Y}}_\mathbf{l} - \mathbf{S}\mathbf{Y}_\mathbf{l} \right\|^2 + \right.$$

$$\left. \mu_1 \sum_{i,j} \mathbf{M}_{ij}(\hat{\mathbf{Y}}_{\mathbf{il}} - \hat{\mathbf{Y}}_{\mathbf{jl}})^2 + \mu_2 \left\| \hat{\mathbf{Y}}_\mathbf{l} - \mathbf{R}_l \right\|^2 \right]$$

Here $\mathbf{M}$ is the symmetrized weight matrix, $\mathbf{Y_{jl}}$ is the initial weight assignment or the seed weight for label $l$ on node $j$, $\hat{\mathbf{Y}}_{\mathbf{jl}}$ is the updated weight of the label $l$ on node $j$. $\mathbf{S}$ is diagonal matrix indicating seed nodes, and $\mathbf{R}_{jl}$ is the regularization target for label $l$ on node $j$. Here, we are assuming a classification task with $K$ labels, and MAD introduces a dummy label as an initial assignment for the unlabeled nodes.

We follow Huang et al. (2021) and perform two post-processing steps. While the original approach use label spreading (Zhou et al., 2003) for both steps, we replace it with MAD. Moreover, our graphs are constructed by a combination of embedding-based similarity and n-gram based similarity as described in §2.3, i.e. $\mathcal{G}_{pred}$. Both the postprocessing steps are applied on the same graph structure. However, the seed label initializations differ in both settings.

**Propagation of Residual Errors:** We obtain the predictions from the base predictor, where each prediction is a distribution over the labels. Using the predictions, we compute the residual errors for the training nodes and propagate the residual errors through the edges of the graph. The unlabelled and validation nodes are initialized with a zero value (or a dummy value), and the seed nodes are initialized with their residuals. Essentially $\mathbf{Y}$ is initialized with a non-zero error for the training nodes with a non-zero residual error. With this initialization of $\mathbf{Y}$ we apply MAD on $\mathcal{G}_{MAD}$. The key assumption here is that the errors in the base prediction are positively correlated with the similarity neighborhood in the graph and hence the residuals need to be propagated (Huang et al., 2021). Here, the residuals are propagated. Hence at the end of the propagation, each node has the smoothed errors as a distribution over the labels. To get the predictions after this step, the smoothed errors are added to predictions from the base predictor for each node.

**Smoothing Label Distribution** The last step in our classification pipeline involves a smoothing step. Here, we make the fundamental assumption of homophily, where adjacent nodes tend to have similar labels. Here, $\mathbf{Y}$ is initialized as follows: Seed labels are provided with their ground truth labels, the validation nodes and the unlabelled nodes are provided with initialized with the predictions after the error propagation step. With this initialization, we perform MAD over $\mathcal{G}_{MAD}$. In multiclass

classification, the label with the maximum value for each node is predicted as the final class. In multilabel classification, all the labels with a score above a threshold are predicted as the final labels.

# 3 Experimental Setup

We perform experiments for the three intent related tasks - Intent Discovery, Multiclass Intent Detection, and Multi-label Intent Detection. Here, we provide training and modeling details that are common to all three tasks and then mention task-specific details such as the baselines and evaluation metrics at appropriate sections.

**Pretraining Datasets.** One feature of IN-TENDD is the unification of these three tasks via a common pretrained transformer backbone. This common pretraining step is performed on CLINC-150 (Larson et al., 2019), BANK-ING77 (Casanueva et al., 2020), HWU64 (Liu et al., 2019a), NLU++ (Casanueva et al., 2022), and StackOverflow (Xu et al., 2015). Following prior work on contrastive learning for intent detection by Zhang et al. (2021c), we additionally include TOP (Gupta et al., 2018), SNIP (Coucke et al., 2018), and ATIS (Tür et al., 2010). Table 4 shows some of the relevant statistics for the datasets.

**Training and Modeling Details.** We choose RoBERTa (Liu et al., 2019b) with the base configuration as our common encoding backbone and pretrain with aforementioned datasets. For encoding the input utterances, we use a cross-encoder architecture as detailed by (Mesgar et al., 2023). In this setup, the joint embedding for any pair of utterances $(p, q)$ –needed for contrastive learning for instance– is obtained by embedding it as "[CLS] p [SEP] q" and the [CLS] representation is used as the representation for that pair. Mesgar et al. (2023) found that a cross-encoder approach works much better than a Bi-encoder where any pair of utterances are independently embedded.

We perform all of our experiments using the `tranformers` library (Wolf et al., 2020) and the `pytorch` framework (Paszke et al., 2019). We train our models using the AdamW optimizer with learning rate set to 2e-5, warmup rate of 0.1, and weight decay of 0.01. We pretrain our model for 15 epochs, and thereafter perform task-specific training for another 20 epochs. All experiments are performed on a machine with NVIDIA A100 80GB and we choose the maximum batch size that

fits the GPU memory ($= 96$). We perform hyperparameter search for the temperature $\tau$ and lambda $\lambda$ over the ranges $\tau \in \{0.1, 0.3, 0.5\}$, and $\lambda \in \{0.01, 0.03, 0.05\}$.

## 4 Experiments and Results

### 4.1 Intent Discovery

**Datasets.** We use three datasets for benchmarking INTENDD for Intent Discovery, namely, BANKING77, CLINC-150, and Stack Overflow. We assess the effectiveness of our proposed ID system in two practical scenarios: unsupervised ID and semi-supervised ID. To ensure clarity, we introduce the term Known Intent Ratio (KIR), which represents the ratio of known intents in the training data: the number of known intent categories ($|\mathcal{I}_k|$) divided by the sum of the known intent categories and unknown categories ($|\mathcal{I}_k| + |\mathcal{I}_u|$). In this context, a value of $|\mathcal{I}_k| = 0$ corresponds to unsupervised ID, indicating the absence of any known intent classes. For semi-supervised ID, we adopt the approach outlined in previous works (Kumar et al., 2022; Zhang et al., 2021b), conducting experiments using three KIR values: $\{25\%, 50\%, 75\%\}$.

**Evaluation Metrics.** Following previous work (Zhang et al., 2021b), we report three metrics, namely Clustering Accuracy (**ACC**) (Yang et al., 2010), Normalized Mutual Information (**NMI**) (Strehl and Ghosh, 2002), Adjusted Rand Index (**ARI**) (Hubert and Arabie, 1985). All metrics range between 0 and 100 and larger values are more desirable.

**Baselines.** We follow the recent work of Kumar et al. (2022) to select suitable baselines for unsupervised and semi-supervised scenarios. Due to space constraints, we detail these in the appendix.

**Results.** We report all the intent discovery results in table 1. To begin with, it is important to highlight that our proposed method INTENDD consistently demonstrates superior performance surpassing all baseline techniques in both unsupervised and semi-supervised settings across all three datasets. Specifically, in an entirely unsupervised scenario, SBERT-KM emerges as the most formidable baseline, yet our approach significantly outperforms it. It should be noted that the fundamental distinction between INTENDD and SBERT-KM lies in our graph construction strategy for clustering. Our strategy relies on a combination of semantic similarity (via embeddings) and n-gram based similarity

(via keyphrases), underscoring the importance of incorporating both these similarity measures.

Furthermore, while our approach demonstrates notable enhancements across all configurations, these improvements are particularly pronounced when the amount of labeled data is limited, resulting in an average increase of nearly 3% in accuracy for KIR values of 0% and 25%.

### 4.2 Multiclass Intent Detection

**Datasets and Evaluation Metric.** Following Zhang et al. (2021c), we perform few-shot intent detection and select three challenging datasets for our experiments, namely, CLINC-150, BANKING77, and HWU64. We use the same training and test splits as specified in that paper, and use detection accuracy as our evaluation metric.

**Baselines.** Due to space constraints, we provide detailed description of all baselines in the appendix (please refer §A.1). We use the following baselines: RoBERTa-base (Zhang et al., 2020), CONVBERT (Mehri et al., 2020), CONVBERT + Combined Mehri and Eric (2021), (Zhang et al., 2020), and CPFT (Zhang et al., 2021c, Contrastive Pre-training and Fine-Tuning). CPFT is the current state-of-the-art employing self-supervised contrastive pre-training on multiple intent detection datasets, followed by fine-tuning using supervised contrastive learning.

**Results.** Table 2 shows the results of our experiments for multiclass intent detection. Our proposal, INTENDD demonstrates superior performance across all three setups when compared to the baseline models in the 5-shot, 10-shot, and full data scenarios. In the 5-shot setting, exhibits an average absolute improvement of 2.47%, with the highest absolute improvement of 4.31% observed in the BANKING77 dataset. Across all the datasets, INTENDD achieves average absolute improvements of 1.31% and 0.71% in the 10-shot and full data settings, respectively.

INTENDD currently does not incorporate any augmented data in its experimental setup. We do not compare our work with data augmentation methods as they are orthogonal to ours. One such example is that of ICDA (Lin et al., 2023), where a large language model (OPT-66B) (Zhang et al., 2022) is used to augment the intent detection datasets for few-shot data settings. Nevertheless, we find that our method performs better than ICDA. We mention this comparison in the appendix B.1.

| | Method | CLINC | | | BANKING | | | STACK OVERFLOW | | |
|---|---|---|---|---|---|---|---|---|---|---|
| | | ACC | NMI | ARI | ACC | NMI | ARI | ACC | NMI | ARI |
| Unsupervised | BERT-KM | 45.06 | 70.89 | 26.86 | 29.55 | 54.57 | 12.18 | 13.85 | 11.60 | 1.60 |
| | DAC | 55.94 | 78.40 | 40.49 | 27.41 | 47.35 | 14.24 | 16.30 | 14.71 | 2.76 |
| | DCN | 49.29 | 75.66 | 31.15 | 41.99 | 67.54 | 26.81 | 57.09 | 61.34 | 34.98 |
| | DEC | 46.89 | 74.83 | 27.46 | 41.29 | 67.78 | 27.21 | 57.09 | 61.32 | 21.17 |
| | SAE-KM | 46.75 | 73.13 | 29.95 | 38.92 | 63.79 | 22.85 | 37.16 | 48.72 | 23.36 |
| | SBERT-KM | 61.04 | 82.22 | 48.56 | 55.72 | 74.68 | 42.77 | - | - | - |
| | INTENDD (Ours) | **63.87** | **83.12** | **51.76** | **58.74** | **75.91** | **47.88** | **79.32** | **73.88** | **62.49** |
| KIR = 25% | CDAC+ | 64.64 | 84.25 | 50.35 | 48.71 | 69.78 | 35.09 | 74.30 | 74.33 | 39.44 |
| | DeepAligned | 73.71 | 88.71 | 64.27 | 48.88 | 70.45 | 36.81 | 69.66 | 70.23 | 53.69 |
| | DSSCCBERT | 75.72 | 89.12 | 66.72 | 55.52 | 72.73 | 42.11 | - | - | - |
| | DSSCCSBERT | 80.36 | 91.43 | 72.83 | 64.93 | **80.17** | 53.60 | 81.72 | 76.57 | 68.00 |
| | INTENDD (Ours) | **83.11** | **92.32** | **76.31** | **67.50** | 76.79 | **57.85** | **84.82** | **78.93** | **71.64** |
| KIR = 50% | CDAC+ | 69.02 | 86.18 | 54.15 | 53.34 | 71.53 | 40.42 | 76.30 | 76.18 | 41.92 |
| | DeepAligned | 80.22 | 91.63 | 72.34 | 59.23 | 76.52 | 47.82 | 72.89 | 74.49 | 57.96 |
| | DSSCCBERT | 81.46 | 91.39 | 73.48 | 63.08 | 77.60 | 50.64 | - | - | - |
| | DSSCCSBERT | 83.49 | 92.78 | 76.80 | 69.38 | 82.68 | 58.95 | 82.43 | 77.30 | 68.94 |
| | INTENDD (Ours) | **84.57** | **93.91** | **78.42** | **71.16** | **84.56** | **63.17** | **85.01** | **79.14** | **72.49** |
| KIR = 75% | CDAC+ | 69.89 | 86.65 | 54.33 | 53.83 | 72.25 | 40.97 | 75.34 | 76.68 | 43.97 |
| | DeepAligned | 86.01 | 94.03 | 79.82 | 64.90 | 79.56 | 53.64 | 74.51 | 76.24 | 59.45 |
| | DSSCCBERT | 87.91 | 93.87 | 81.09 | 69.82 | 81.24 | 58.09 | - | - | - |
| | DSSCCSBERT | 88.47 | 94.50 | 82.40 | 75.15 | 85.04 | 64.83 | 82.65 | 77.08 | 68.67 |
| | INTENDD (Ours) | **90.99** | **96.29** | **83.62** | **77.08** | **87.39** | **68.69** | **85.47** | **77.12** | **72.90** |

Table 1: **Results for Intent Discovery.** First set of results are in a completely unsupervised setting, while others are when some of the intent categories are known. KIR is used to represent the Known Intent Ratio. In all the experiments involving known intents classes, we assume the proportion of labeled examples to be 10% (Kumar et al., 2022). Baseline results are taken from Kumar et al. (2022) and those marked with - have not been reported in literature. DSSCC paper does not report results for DSSCCBERT on Stack Overflow, and we could not get access to their code to independently run that model. The best results for each dataset and setting are marked in **bold**. We note that our proposed method consistently outperform recent baselines by a significant margin.

| Method | BANKING77 | | | HWU64 | | | CLINC150 | | |
|---|---|---|---|---|---|---|---|---|---|
| | 5 | 10 | Full | 5 | 10 | Full | 5 | 10 | Full |
| RoBERTa | 74.65 | 84.67 | 93.08 | 76.75 | 83.42 | 90.97 | 88.27 | 91.21 | 96.46 |
| CONVBERT | - | 83.63 | 92.95 | - | 83.77 | 90.43 | - | 92.10 | 97.07 |
| + MLM | - | 83.99 | 93.44 | - | 84.52 | 92.38 | - | 92.75 | 97.11 |
| + MLM + Example | - | 84.09 | 94.06 | - | 83.44 | 92.47 | - | 92.35 | 97.11 |
| + Combined | - | 85.95 | 93.83 | - | 86.28 | 93.03 | - | 97.97 | 97.31 |
| DNNC | 80.40 | 86.71 | - | 80.46 | 84.72 | - | 91.02 | 93.76 | - |
| CPFT | 80.86 | 87.20 | - | 82.03 | 87.13 | - | 92.34 | 94.18 | - |
| INTENDD-MLP (Ours) | 82.17 | 88.70 | 93.63 | 81.27 | 85.32 | 92.89 | 91.34 | 93.66 | 96.92 |
| INTENDD-EP (Ours) | 83.25 | 88.96 | 94.18 | 83.17 | 86.35 | 93.31 | 92.70 | 92.24 | 97.93 |
| INTENDD (Ours) | **85.34** | **89.62** | **94.86** | **84.11** | **88.37** | **93.64** | **93.52** | **94.71** | **98.03** |

Table 2: **Results for Multiclass Intent Detection.** We report intent detection accuracy for three data settings. We use the baseline numbers from (Lin et al., 2023). The best results for each dataset and setting are marked in **bold**.

**Is Modified Adsorption important for Intent Detection?** INTENDD uses a pipeline of three classification setups: one using the MLP, and two in a transductive setting using the Modified Ad-

sorption (MAD). We perform ablation experiments with these components and report results in the table 2. We report results from three systems by progressively adding one component at a time. IN-TENDD-MLP denotes the results without using the two steps of Modified Adsorption, INTENDD-EP denotes the results with MAD but only the residual propagation step (i.e. without the label smoothing). We observe consistent performance improvements due to each of the components of the pipeline. Notably, the label propagation step leads to more significant improvements and these gains are not only observed in the few-shot setups but also in the fully data scenarios.

### 4.3 Multilabel Intent Detection

**Datasets and Evaluation Metric.** Following Vulić et al. (2022), we use three datasets for multilabel intent detection: BANKING77, Mix-ATIS, and HOTELS subset is taken from NLU++ benchmark. MixATIS consists of a multilabel dataset synthetically obtained via concatenating single-label instances from the ATIS dataset. We do not perform experiments with InsuranceFAQ from that paper since it was an internal data. We report standard evaluation metrics: F1 and exact match accuracy (Acc). We report results on all datasets in two settings: *low-data*, and the *high-data* regimes, again replicating the experimental settings from Vulić et al. (2022).

**Baselines.** Our main baseline is the MultiConvFiT model proposed by Vulić et al. (2022) with two variants. MultiConvFiT (FT) where full fine-tuning along with the updating encoder parameters is performed. The second, more efficient alternative MultiConvFiT (Ad) where an adapter is used instead of updating all parameters. Along with this, two other baselines from ConVFiT (Vulić et al., 2021) are adapted –DRoB, and mini-LM. Please refer to Vulić et al. (2022) for more details on these methods.

**Results.** The results of our experiments are shown in table 3. First, the results demonstrate consistent gains achieved by our method across all three datasets. Notably, in low-data scenarios, we observe an average increase of approximately 1% in F-scores. As anticipated, the performance enhancements are more substantial in low-data settings. However, it is noteworthy that our model outperforms MultiConVFiT even in high-data setup.

We find the results of our base predictor and our final classifier to be statistically significant for all the settings of multi-class and multi-label intent detection using the t-test ($p < 0.05$).

## 5 Conclusion

In summary, this paper presents a novel approach, INTENDD, for intent detection and discovery in task-oriented dialogue systems. By leveraging a shared utterance encoding backbone, IN-TENDD unifies intent classification and novel intent discovery tasks. Through unsupervised contrastive learning, the proposed approach learns representations by generating pseudo-labels based on lexical features of unlabeled utterances. Additionally, the paper introduces a two-step post-processing setup using modified adsorption for classification tasks. While intent classification tasks typically focus on contrastive representation learning or data augmentation, we show that a two-step post-processing setup in a transductive setting leads to statistically significant improvements to our base classifier, often rivaling or at par with data augmentation approaches. Extensive evaluations on diverse benchmark datasets demonstrate the consistent improvements achieved by our system over competitive baselines.

## 6 Limitations

While our research provides valuable insights and contributions, we acknowledge certain limitations that should be considered. In this section, we discuss two main limitations that arise from our work.

First, a limitation of our proposed intent discovery algorithm is its reliance on prior knowledge of the number of intent clusters. This assumption may not hold in real-world scenarios where the underlying intent structure is unknown or may change dynamically. The requirement of knowing the exact number of intent clusters can be impractical and unrealistic, limiting the generalizability of our approach. However, we recognize that this limitation can be addressed through modifications to our algorithm. Future investigations should explore techniques that allow for automated or adaptive determination of the number of intent clusters, making the approach more robust and applicable to diverse real-world settings.

The second limitation of our research lies in the reliance on the construction of a graph using extracted keyphrases during the contrastive pretrain-

| Method | BANKING77 | | HOTELS | | MIXATIS | |
| --- | --- | --- | --- | --- | --- | --- |
| | *low-data* | *high-data* | *low-data* | *high-data* | *low-data* | *high-data* |
| DRoB (Vulić et al., 2021) | 70.6 / 31.0 | 86.7 / 60.0 | 65.3 / 46.8 | 80.9 / 65.1 | 58.5 / 21.2 | 78.4 / 46.9 |
| mini-LM (Vulić et al., 2021) | 70.8 / 31.2 | 86.7 / 58.1 | 64.2 / 46.0 | 80.3 / 65.8 | 58.1 / 22.0 | 78.6 / 47.5 |
| MultiConvFiT (Ad) | 80.7 / 47.9 | 93.7 / 77.2 | 67.3 / 47.6 | 92.8 / 84.9 | 73.7 / 44.6 | 90.8 / 78.3 |
| MultiConvFiT (FT) | 81.9 / 49.1 | 94.3 / 80.5 | 70.2 / 51.1 | 93.4 / 84.0 | 76.5 / 51.4 | 91.5 / 81.1 |
| INTENDD(Ours) | **82.4 / 49.7** | **94.8 / 80.9** | **71.5 / 51.8** | **93.7 / 84.3** | **77.6 / 51.9** | **91.9 / 81.5** |

Table 3: **Results for Mult-label Intent Detection.** We report both $F_1$ score and Accuracy for all the settings. The first number in each cell is the $F_1$ score and the second number is the accuracy. Vulić et al. (2022) proposed two variants for MultiConvFiT - one with full fine-tuning (FT) and another with adapters (Ad). The ConvFiT model proposed by Vulić et al. (2021) has been adapted for multi-label settings with DistilRoBERTa (DRoB), and mini-LM as backbones. The best results for each dataset and setting are marked in **bold**. To establish the statistical significance of our results, we performed the paired t-test between INTENDD and MultiConvFiT (FT) and found the p-value in all cases to be $< 0.05$.

ing step, which is a common requirement across all three tasks explored in our study. While this graph construction step facilitates the representation learning process, it introduces a constraint on the flexibility of modifying the graph structure. Even a minor modification to the graph construction would necessitate retraining all systems, which can be time-consuming and resource-intensive. Currently, we mitigate the need for covering new utterances (with no overlapping keyphrases) by simply relying on similarity from the encoder representation itself. However, it still may still lead to concept drift over time, and the representation might need to be updated by retraining all the modules in INTENDD. In future work, we intend to explore alternative approaches that offer more flexibility in graph construction, allowing for easier modifications without the need for extensive retraining. By addressing this limitation, we aim to enhance the adaptability and scalability of our framework.

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

| Name | $|\mathcal{D}|$ | #Intent | Tasks |
|---|---|---|---|
| CLINC-150 | 18,200 | 150 | MC, ID |
| BANKING77 | 10,162 | 77 | MC, ID |
| HWU64 | 10,030 | 64 | MC |
| NLU++ | 3,080 | 62 | ML |
| MIXATIS | 20,000 | 18 | ML |
| STACK OVERFLOW | 20,000 | 20 | ID |

Table 4: **Dataset Statistics** for the three Intent Identification tasks explored in this work. Second column lists the number of intent classes in each of the datasets. Key: MC - Multiclass (Single label), ML - Multi-label, ID - Intent Discovery.

| | IntenDD | | CPFT | |
|---|---|---|---|---|
| | 5 | 10 | 5 | 10 |
| BANKING77 | 0.38 | **0.29** | **0.20** | 0.48 |
| HWU | **0.35** | **0.18** | 0.51 | 0.25 |
| CLINC150 | **0.32** | 0.21 | 0.39 | **0.18** |

Table 5: Standard Deviation across different runs for Few-Shot Intent Detection. We observe that, compared to CPFT, our method has lower variance across most settings.

*Conference on Machine Learning*, ICML'03, page 912–919. AAAI Press.

## A  Experimental Details

### A.1  Baseline Description

**Intent Discovery**   In the unsupervised setting, the first two baselines use K-means algorithm (MacQueen, 1967) on top of sentence embeddings from BERT (Devlin et al., 2019) and SBERT (Reimers and Gurevych, 2019) to cluster user utterances (**BERT-KM**, **SBERT-KM** respectively). **DEC** (Xie et al., 2016) is a two step deep clustering approach involving a Stacked Autoencoder (SAE) along with confidence based cluster assignment. **SAE-KM** uses K-means with SAE (Xie et al., 2016), **DCN** (Yang et al., 2017) is a method that performs dimensionality reduction and clustering using a joint objective function, and **DAC** (Chang et al., 2017) treats the clustering problem as a pairwise binary classification problem to learn cluster centers.

For the semi-supervised case, we use **CDAC+** (Lin et al., 2020), in which the pairwise constraints from the labeled examples are incorporated into the clustering problem. **DeepAligned** (Zhang et al., 2021b) uses labeled data to generated pseudo labels as well as pretrain a BERT model followed by K-means clustering. Finally, we compare our method with a very recent method **DSCC** (Kumar et al., 2022) where the authors propose an end-to-end contrastive clustering algorithm to jointly learn cluster centers and utterance representations via a combination of supervised and self-supervised methods. We report results with two backbone models used in the paper, BERT and S-BERT.

**Multiclass Intent Detection**   In this study, we consider several baseline models for intent de-

tection. The first baseline, RoBERTa-base, utilizes **RoBERTa** as its base model, supplemented with a linear classifier on top for classification purposes. Another baseline, **CONVBERT**, involves fine-tuning BERT using a vast open-domain dialogue corpus consisting of 700 million conversations (Mehri et al., 2020). Furthermore, **CONVBERT + Combined**, an intent detection model based on CONVBERT, adopts example-driven training with similarity matching and transformer attention observers, along with task-adaptive self-supervised learning using masked language modeling on intent detection datasets. The term "Combined" refers to the optimal MLM+Example+Observers setting described in Mehri and Eric (2021). Another baseline model, **DNNC** (Discriminative Nearest-Neighbor Classification) (Zhang et al., 2020), employs a discriminative nearest-neighbor approach, matching training examples based on similarity and employing data augmentation during training. Additionally, it enhances performance through pre-training on three natural language inference tasks. Finally, **CPFT** (Contrastive Pre-training and Fine-Tuning) (Zhang et al., 2021c) represents the current state-of-the-art in few-shot intent detection, employing self-supervised contrastive pre-training on multiple intent detection datasets, followed by fine-tuning using supervised contrastive learning.

## B  Additional Results

**Variance in Few-shot Intent Detection.**   In the few-shot settings, we generally report lower variance than CPFT, the system with the second-best results consistently. Table 5 shows the standard deviation for INTENDD and CFPT, where CPFT has a lower variance than INTENDDonly in two out of six settings.

### B.1 Comparisong with a recent data augmentation strategy - ICDA

INTENDD currently does not incorporate any augmented data in its experimental setup. We do not compare our work with data augmentation methods as they are orthogonal to ours. One such example is that of ICDA (Lin et al., 2023), where a large language model (OPT-66B) (Zhang et al., 2022) is used to augment the intent detection datasets for few-shot data settings. Nevertheless, we find that our method performs better than ICDA.

INTENDD outperforms all the settings of ICDA in both 5-shot and Full data settings. In 10-shot settings, while INTENDDreports the best results on HWU64, the largest configurations of ICDA reports a better accuracy for the other two datasets. The largest configuration uses 128 times more augmented data than the available supervised data to report the best results. Overall, ICDA reports an accuracy of 89.79 % and 94.84 % on BANKING77 and CLINC150 respectively which is 0.17 % and 0.13 % more than INTENDD.

### B.2 Computing Infrastructure Used

All of our experiments required access to GPU accelerators. We ran our experiments on three machines: Nvidia Tesla A100 (80 GB VRAM), Nvidia Tesla V100 (16 GB VRAM), Tesla A100 (40 GB VRAM).