# OpenReview forum: "IntenDD: A Unified Contrastive Learning Approach for Intent Detection and Discovery"
_EMNLP/2023/Conference — EMNLP 2023 Findings_

### Official Review · Reviewer_WzPH · 2023-08-07

**Typos Grammar Style And Presentation Improvements:** 1. It is advisable to add a figure of…
**Soundness:** 3

**Excitement:**

4: Strong: This paper deepens the understanding of some phenomenon or lowers the barriers to an existing research direction.

**Missing References:**

Zhang, Hanlei, et al. "TEXTOIR: An Integrated and Visualized Platform for Text Open Intent Recognition." Proceedings of the 59th Annual Meeting of the Association for Computational Linguistics and the 11th International Joint Conference on Natural Language Processing: System Demonstrations. 2021.

**Paper Topic And Main Contributions:**

This paper is centered around INTENDD, a novel and unified contrastive learning approach for intent detection and discovery in task-oriented dialogue systems.
The primary contributions include (1) the formulation of INTENDD that unifies the tasks of intent classification and discovery using a shared utterance encoding backbone, (2) an entirely unsupervised contrastive learning strategy for representation learning, and (3) a two-step post-processing setup using modified adsorption(variant of label propagation) for intent classification. The study also presents performance improvements in comparison with other baselines across several benchmark datasets.


**Questions For The Authors:**

1. About motivation, what is the intuition to unified tasks by introducing graphs with Louvain and MAD?  If I used K-means with representations learned by INTENDD, will it also work well?
2. Significant tests are needed to prove the effectiveness, such as a single-tail t-test. From Table 2, adding the MAD increases merely about 1% in accuracy;
3. Does the model require a separate graph to be constructed for each dataset?
4. What are the time and space complexities associated with the construction of these graphs?
5. In Table 4, Is the continued pre-training / graph constructed using the entire dataset or only the training set?

**Reasons To Accept:**

1. Novel unified approach to intent detection (multi-class and multi-label) and discovery (unsup. and semi-sup).
2. The introduction of the two-step post-processing method shows performance improvement, demonstrating the efficacy of the proposed model.
3. The extensive experiments across various datasets and tasks makes it a valuable contribution to the research community.

**Reasons To Reject:**

1. Some important details is unclear and should be clarified. For example: (1) needs to summarize the total loss function of each stage/task, including how to balance the loss. (2) The final prediction for intent classification (multi-class/label) and intent discovery (unsup., semi-sup.) should be formally defined in the formula.
2. The absence of a comprehensive discussion of the construction of the graph structure in the transductive learning setting might be a concern for readers aiming to replicate or extend the model.
3. The proposed method necessitates corpus-specific representation learning. It does not follow the pattern of unified training followed by direct inference across multiple tasks and datasets.

**Reproducibility:**

3: Could reproduce the results with some difficulty. The settings of parameters are underspecified or subjectively determined; the training/evaluation data are not widely available.

**Reviewer Confidence:**

3: Pretty sure, but there's a chance I missed something. Although I have a good feel for this area in general, I did not carefully check the paper's details, e.g., the math, experimental design, or novelty.

---

> ### Author Rebuttal · Authors · 2023-08-29
>
> We thank the reviewer for their time and the thoughtful review.
>
> **Loss functions used**
>
> Thank you for pointing this out. Due to space constraints, we had to omit the formulae for loss functions, but we will add this in the final version.
>
> **Graph Construction Procedure**
> We will add a pictorial illustration to clarify the graph construction procedure in the appendix of the paper to improve the readability.
>
> **Re: Corpus-specific representation learning**
>
> First, we want to emphasize that by unifying we simply mean that we are the first ones to propose a shared encoding backbone for the three tasks of intent detection and discovery. Prior works  in literature, such as the baselines we consider, use different backbones with different pre-training procedures for each of these tasks
>
> **Re: Missing Reference**
>
> Thank you for pointing this out. We will add this to the final version.
>
> **For Q1:**  Good point. First, our motivation to unify the tasks is an important practical concern in the industry where separate pre-training procedures for each of the three tasks are undesirable and hard to maintain. We use Louvain and MAD since these are well-known graph construction methods (ex: for community detection). Using Louvain, we are able to additionally incorporate valuable lexical semantics-level information which is often not explicitly captured in contrastive representation learning approaches.  Hence it complements the information captured in contrastive learning techniques. Using MAD, we apply both error propagation (EP) and smoothing. Here, EP propagates residual errors in training data to correct errors in test data and the smoothing step enables to penalize similar models that are assigned with different labels, or more technically it smoothes the prediction in the test data based on the similarity between data points. In our few shot (5-shot) multiclass settings, the use of MAD has resulted in at least 2 % absolute improvements for all the three datasets we experimented with.
>
>  We expect our method to work better than k-means since it involves a two-step process of label propagation and label smoothing (as part of MAD).
>
> **For Q2**: Indeed, we performed the paired t-test to verify the effectiveness of our method (p-value in all cases was < 0.05). We will mention this in the paper.
> **Regarding  MAD**: As compared to IntenDD-MLP, our configuration where MAD is not employed,  IntenDD uses a two-step pipeline application of MAD (error propagation step and smoothing step). In few-shot settings (5-shot), IntenDD has been shown to increase the performance by at least 2 % absolute improvement for all the datasets as compared to IntenDD-MLP, and the maximum being 3.17 % absolute improvement for Banking. We achieve statistically significant gains in all the cases in Table 2 (10-shot and full), though the gains are merely 1% for full shot setting as highlighted by you. Given that, intent classes can be quite high and can be highly dependent on a business use-case, few shot settings are desirable for intent-related tasks in ToD systems. Hence, we believe the gains by MAD in few shot settings is a valuable addition to the pipeline.
>
> **For Q3:** Yes, the model requires constructing a separate graph for each dataset. We follow this from previous works (like: Vulic et al. 2022 (https://aclanthology.org/2022.emnlp-main.512.pdf )) that use dataset specific procedures after common pre-training.
>
>
> **For Q4:** For graph construction, the nodes remain constant and the edges are constructed by using a feature selection approach. For each iteration of the feature selection approach, we perform the Louvain algorithm. A single run of the Louvain algorithm has a complexity of $O(n.log n)$, where $n$ is the number of nodes. It is known to be a fast community Detection approach. We run that for $d$ iterations in the worst case, where $d$ is the number of features. Altogether, the time complexity for graph construction is $O(n.d.log n)$ . Now, The aforementioned time complexity is comparable to that of the K-Nearest neighbors (K-NN) algorithm. For K-NN the complexity is  $O(n.d.k)$, or a $O(nd + kn)$ for a faster version, where $k$ is the number of neighbors.  Moreover, the largest dataset we currently consider includes 20,000 nodes where a single run of Louvain is often completed in less than 10 seconds in an Intel i-5 CPU and 16 GB RAM, in terms of the wall clock time. Please note that in case of larger network sizes, say those with $10^7$ nodes ore more, we may alternatively use the Leiden algorithm to improve upon the speed. However, that comparison does not fall within the scope of the current work. Further, obtaining and scoring keyphrases does not take more than 14 minutes of wall clock time in a CPU with the aforementioned configuration. This method requires one pass of the Dataset and the set of keyphrases (P)  for collecting n-gram counts of keyphrases and another for calculating the PMI scores.  For larger datasets, we may even use parallelization that can substantially reduce the wall clock time computation.
>
> **For Q5**: Using only the training set.

---

### Official Review · Reviewer_DQWw · 2023-08-10

**Soundness:** 3

**Excitement:**

3: Ambivalent: It has merits (e.g., it reports state-of-the-art results, the idea is nice), but there are key weaknesses (e.g., it describes incremental work), and it can significantly benefit from another round of revision. However, I won't object to accepting it if my co-reviewers champion it.

**Paper Topic And Main Contributions:**

This paper explores the detection and discovery of intents within dialogue utterances in task-oriented dialogue systems. It aims to unify different intent-related tasks, including multiclass and multilabel intent classification, as well as intent discovery in unsupervised and semi-supervised settings.

Contribution 1: IntenDD is introduced, combining intent detection and discovery through a shared utterance encoding backbone. The model leverages an unsupervised contrastive learning method for representation learning and uses pseudo-labeling for unlabeled utterances based on n-gram lexical features.

Contribution 2: A two-step post-processing setup is incorporated for classification tasks using Modified Adsorption (MAD). This process involves propagating residuals in the training data and smoothing the labels in a transductive setting.

Contribution 3: The paper highlights the application of classical transductive learning approaches and how the construction of a graph structure in a transductive setting might contribute to improving model performance.

Contribution 4: The evaluations presented in this work show that the proposed approach performs better than competitive baselines across various tasks under the selected evaluation metrics.

**Reasons To Accept:**

Strength 1: The experimentation is thorough, compares to strong baselines and the paper performs a good analysis.

Strength 2: Detailed explanation of the approach.


**Reasons To Reject:**

W1: In section 2.2, the authors describe the process of using PMI to identify keyphrases representative of a target corpus for representation learning. However, there are several limitations regarding this approach:

1. The process includes row-normalizing the weight matrix W. However, it is not clear what specific normalization technique is used, or how it might affect the properties of the graph or the success of the community detection.

2. The calculation of PMI includes additional terms to avoid bias toward rare words and to favor longer n-grams. These modifications might address some of the common problems with PMI but might introduce new biases or sensitivities. For instance, the square of the number of words in the n-gram could heavily bias the system towards very long phrases, which might or might not be appropriate depending on the nature of the corpus.

3. Depending on the size of D and P, the graph G_D may be quite large, and constructing and analyzing it may be computationally expensive.

4. The calculation of PMI includes additional terms to avoid bias toward rare words and to favor longer n-grams. These modifications might address some of the common problems with PMI but might introduce new biases or sensitivities. For instance, the square of the number of words in the n-gram could heavily bias the system towards very long phrases, which might or might not be appropriate depending on the nature of the corpus.

W2: In section 2.3, the similarity matrix A is constructed based on the cosine similarity between utterance encoding of 2 nodes. However, if the underlying representations (e.g., embeddings) are high-dimensional, computing cosine similarities can be computationally expensive. Also, when creating a new graph G_pred using the weighted average between W and A, the process of conducting weighted averaging is missing. If the weighted averaging process is not thoughtfully designed, it could lead to loss of discriminative information.

W3: In section 2.4,  as a transductive learning method, MAD requires access to all labeled and unlabeled data upfront. It may not generalize well to unseen data or new instances that weren't included in the original graph. Also, MAD introduces a dummy label for the unlabeled nodes. If not managed carefully, this can sometimes lead to ambiguity or confusion in the classification, especially if there's a large imbalance between labeled and unlabeled instances.

W4: For evaluation on intent classification, more evaluations are required on other datasets including MixSNIPS, and FSPS. Also, compared with previous works' multi-label classification results on different benchmark datasets, IntenDD does not have significant improvements from the perspectives of F1 and accuracy shown in Table 3. The author may need to re-examine their approaches and the implementation part.


**Reproducibility:**

3: Could reproduce the results with some difficulty. The settings of parameters are underspecified or subjectively determined; the training/evaluation data are not widely available.

**Reviewer Confidence:**

5: Positive that my evaluation is correct. I read the paper very carefully and I am very familiar with related work.

**Typos Grammar Style And Presentation Improvements:**

This paper is very hard to follow and read. Many paragraphs are repeated and incoherent with each other. For example, when presenting section 2.2, the authors should present the Louvain method and Keyphrase selection for constructing G_D first for avoiding the reader's confusion.

There are also many inconsistent use of abbreviations and acronyms in this paper. For example,  the terms like "ID" (for Intent Discovery), "KIR" (Known Intent Ratio) [line 461- 470], and others are used without clear definitions in the preceding text. This could lead to confusion.

Finally, the proposed approach has a very complex design and needs to be visualized for the reader to better understand how each component works and is connected to each other. Otherwise, it would be too hard for the reader to follow.

---

> ### Author Rebuttal · Authors · 2023-08-29
>
> We thank the reviewer for their time and thoughtful review.
>
> **Re: PMI for identifying keyphrases:**
> 1. We performed min-max normalization for each row – a form of feature scaling as a preprocessing step. For a given node, we wanted to capture the relative importance of other nodes in the graph and hence performed row-wise min-max normalization. During our initial experiments, we observed that we get better partitions when using normalized values over unnormalized values (no test data was used during these initial experiments). We will incorporate the aforementioned details in the draft to avoid any possible ambiguity.
> 2. In our current approach, we used ngrams of up to 5-gram for all the datasets we reported. Moreover, we included an extended-lesk style approach, i.e. multiplying the square of the number of words in a phrase with its PMI, to favor longer keyphrases. In our initial experiments, we observed that we obtained better performance across all the datasets reported in this work when using keyphrases of up to 5-grams and the extended lesk style product (validated on development data).  That said, for the keyphrase scoring approach, we acknowledge that the design decisions may vary for new datasets depending on the nature of those corpora and need to be finalized empirically. Moreover,  the final set of keyphrases is decided not merely using these scores but also based on a feature selection approach (Lines 259-286). Our focus, in this work, is to show that a simple graph construction strategy based on lexical features works well across multiple datasets and we use PMI based scoring approach to capture this intuition. We will add the aforementioned details in the draft to avoid any possible ambiguity.
> 3. Good point. In fact, one of the reasons to use counts of keyphrases is to ensure the relative efficiency of the graph construction procedure. This method requires one pass of D and P for collecting n-gram counts of keyphrases and another for calculating the PMI scores. For graph construction, the nodes remain constant and the edges are constructed by using a feature selection approach. For each iteration of the feature selection approach, we perform the Louvain algorithm. A single run of the Louvain algorithm has a complexity of $O(n.log n)$, where $n$ is the number of nodes. It is known to be a fast community Detection approach. We run that for $d$ iterations in the worst case, where $d$ is the number of features. Altogether, the time complexity for graph construction is $O(n.d.log n)$ . Now, The aforementioned time complexity is comparable to that of the K-Nearest neighbors (K-NN) algorithm. For K-NN the complexity is  $O(n.d.k)$, or a $O(nd + kn)$ for a faster version, where $k$ is the number of neighbors.  Moreover, the largest dataset we currently consider includes 20,000 nodes where a single run of Louvain is often completed in less than 10 seconds in an Intel i-5 CPU and 16 GB RAM, in terms of the wall clock time. Please note that in case of larger network sizes, say those with $10^7$ nodes ore more, we may alternatively use the Leiden algorithm to improve upon the speed. However, that comparison does not fall within the scope of the current work. Further, obtaining and scoring keyphrases does not take more than 14 minutes of wall clock time in a CPU with the aforementioned configuration. For larger datasets, we may even use parallelization that can substantially reduce the wall clock time computation.
> 4. Duplicate of Point 2.
>
> **Constructing similarity matrix A**
>
> The dimension of the utterance encoding for a node in the graph was 768. As previously mentioned, the time complexity for our approach is still close to that of a k-NN-based approach, which is widely used in Intent-related tasks in ToD systems.  Modern scientific libraries like pytorch and scipy handle such cases with little to no difficulty.
>
> Thanks for pointing out the need for additional information on weighted averaging. We obtained the weights via parameter search (grid search) and selected the configuration that optimizes the silhouette score, an intrinsic measure for clustering quality.  We will add the necessary details regarding our weighted average process in the draft.
>
> **Limitation in case of new instances**
>
> Great point. As we acknowledge in the Limitations (L645-651), MAD is a transductive learning approach. Here, we need to update the graph to add new instances which requires re-training the system. As far as generalization is concerned, the neighbors for a newly inserted node (new instance) are determined using the embedding similarity and keyphrae-based overlap. Like in other works on intent classification, we employ contrastive representation learning to learn our embedding. Hence, unseen instances can still leverage their embedding level similarity with instances used during the training for MAD. As stated in lines 655-657, our approach may lead to concept drift over time, and the representation might need to be updated by retraining.
>
> **Re: Evaluation**
>
> To be consistent with the baselines, we choose our datasets from the recent works for the three tasks we considered: Kumar et al. (2022) for Intent Discovery, Lin et al. 2023 for Intent Detection, and Vulic et al. 2022 for multi-label ID.
>
> Regarding the results in Table 3, we performed the paired t-test to establish the significance of results (p-value in all cases was < 0.05). We will mention this in the caption of table 3.

---

### Official Review · Reviewer_MW7X · 2023-08-22

**Soundness:** 3

**Excitement:**

3: Ambivalent: It has merits (e.g., it reports state-of-the-art results, the idea is nice), but there are key weaknesses (e.g., it describes incremental work), and it can significantly benefit from another round of revision. However, I won't object to accepting it if my co-reviewers champion it.

**Missing References:**

The paper should

**Paper Topic And Main Contributions:**

This paper presents IntenDD as a unified modeling approach for both intent discovery/clustering and intent detection/classification tasks. Both tasks in IntenDD share a common continually pretrained utterance encoder that is trained with the combination of contrastive and MLM losses, and then updated using utterances from the target dataset.  Further post-processing in the classification model through transductive learning on a graph structure. Evaluation on intent discovery and multiclass/multilabel intent detection benchmark tasks shows new state-of-the-art accuracy.

**Questions For The Authors:**

* Please clarify if the intent discovery baselines were also evaluated under the "known number of intent classes" assumption

* Please elaborate on the graph construction complexity.

**Reasons To Accept:**

* The paper introduces several novel ideas, especially around the construction of graph structures for pseudo label assignment, and applying Modified Adsorption -- a graph-based semi-supervised transductive learning approach as post-processing of intent classification.

* Empirical evaluation were carried out on multiple datasets to benchmark intent clustering and classification performance of IntenDD.

**Reasons To Reject:**

* A key limitation of the work is the assumption of known number of intent classes during graph construction. Especially for the intent discovery evaluation, I am not sure if the IntenDD experiments reported in Table 1 are all under this assumption, and can be directly comparable to the cited baselines. While the authors acknowledged this as a limitation of the work, it would require further clarification on the intent discovery experiment setting to see if the reported SOTA is in fair comparison with baselines.

* Graph construction procedure described in section 2.2 seems to be very compute-intensive and difficult to scale with the size of dataset D.

**Reproducibility:**

3: Could reproduce the results with some difficulty. The settings of parameters are underspecified or subjectively determined; the training/evaluation data are not widely available.

**Reviewer Confidence:**

5: Positive that my evaluation is correct. I read the paper very carefully and I am very familiar with related work.

**Typos Grammar Style And Presentation Improvements:**

p1. L66. aA -> a
p2. L166. form a connected a weighted directed graph -> form a connected weighted directed graph
p6. L514. zhang-etal-2021-shot <- reference format
p6. L525. zhang-etal-2020-discriminative <- reference format
p7. L570. fully data setups -> full data scenarios
p8. L579. AXIS -> ATIS

---

> ### Author Rebuttal · Authors · 2023-08-28
>
> We thank the reviewer for their time and the thoughtful review.
>
> **Q- "Please clarify if the intent discovery baselines were also evaluated under the "known number of intent classes" assumption"**
>
> **Response:** As mentioned in L474-477 of the submitted draft, we follow the exact same setting as Kumar et al. (2022), the current state of the art in intent discovery (without data augmentation). The same setting is followed by several other recent works such as Zhang et al., (2021a).
>
> We primarily consider three different values for the Known intent ratio (25%, 50%, 75%) and each of them has 10 % of its samples labeled. For the unsupervised setup, no labeled information is used. Further, we assume the total number of intent classes is known apriori. This is a standard evaluation setup for intent discovery. For instance, *Zhang et al. (2022)*, Kumar et al. (2022), and  Zhang et al., (2021a) follow the same setup. In Kumar et al. (2022), they consider two scenarios. In the second scenario, they do not assume that the total number of intent classes is known beforehand. However, we have considered the results based on a comparable setting with ours, where the total number of classes is known beforehand.  Similarly, if we consider *Zhang et al. (2022)*, their experiments are solely based on intent discovery and they only consider the setting where the total number of classes is known beforehand. Since *Zhang et al. (2022)* use data augmentation, it is orthogonal to ours (please refer to lines 544 to 553) and hence we do not compare it with our approach.
>
> Zhang et al. (2022) - https://aclanthology.org/2022.acl-long.21.pdf
>
> **"Please elaborate on the graph construction complexity."**
>
> For graph construction, the nodes remain constant and the edges are constructed by using a feature selection approach. For each iteration of the feature selection approach, we perform the Louvain algorithm. A single run of the Louvain algorithm has a complexity of $O(n.log n)$, where $n$ is the number of nodes. It is known to be a fast community Detection approach. We run that for $d$ iterations in the worst case, where $d$ is the number of features. Altogether, the time complexity for graph construction is $O(n.d.log n)$ . Now, The aforementioned time complexity is comparable to that of the K-Nearest neighbors (K-NN) algorithm. For K-NN the complexity is  $O(n.d.k)$, or a $O(nd + kn)$ for a faster version, where $k$ is the number of neighbors.  Moreover, the largest dataset we currently consider includes 20,000 nodes where a single run of Louvain is often completed in less than 10 seconds in an Intel i-5 CPU and 16 GB Ram, in terms of the wall clock time. Please note that in case of larger network sizes, say those with $10^7$ nodes ore more, we may alternatively use the Leiden algorithm to improve upon the speed. However, that comparison does not fall within the scope of the current work.
>
> We will add these details to the paper.
>
> Leiden - https://www.nature.com/articles/s41598-019-41695-z

---

### Official Review · Reviewer_cGUN · 2023-08-22

**Typos Grammar Style And Presentation Improvements:** 1. line 66
**Soundness:** 3

**Excitement:**

2: Mediocre: This paper makes marginal contributions (vs non-contemporaneous work), so I would rather not see it in the conference.

**Paper Topic And Main Contributions:**

This paper uses an unsupervised contrastive learning strategy for representation learning, where pseudo labels for the unlabeled utterances are generated based on lexical features. This paper also introduces a two-step post-processing setup for the classification tasks using modified adsorption. The residuals in the training data are propagated followed by smoothing the labels both modeled in a transductive setting.

**Questions For The Authors:**

1. Statistics analysis to support the hypothesis that every keyphrase is present at least five times in D
2. More detailed analysis on why the proposed method fails to outperform DSSCCSBERT in BANKING and in KIR=25% setting

**Reasons To Accept:**

1. The method is well-defined and well address the problems. The paper introduces novel ideas about the construction of graph structures for pseudo label assignment and the method Modified Adsorption to perform graph-based semi-supervised transductive learning approach as post-processing of intent classification.
2. Conduct comprehensive evaluations
3. Evaluations on various benchmark datasets prove that the proposed method consistently outperforms competitive baselines across all three tasks.

**Reasons To Reject:**

1. More evaluations for backbones other than RoBERTa is necessary such as BERT to prove the general improvement of the method for most pre-trained models.
2. More evaluations for backbones of more size is preferred.

**Reproducibility:**

2: Would be hard pressed to reproduce the results. The contribution depends on data that are simply not available outside the author's institution or consortium; not enough details are provided.

**Reviewer Confidence:**

3: Pretty sure, but there's a chance I missed something. Although I have a good feel for this area in general, I did not carefully check the paper's details, e.g., the math, experimental design, or novelty.

---

> ### Author Rebuttal · Authors · 2023-08-28
>
> We thank the reviewer for their time and the thoughtful review.
>
> **Q- "Statistics analysis to support the hypothesis that every keyphrase is present at least five times in D"**
> **Response:** As stated in lines 189-191, our choice to incorporate key phrases with a minimum support of five from 𝒟 is a deliberate design choice rather than a hypothesis. This selection also serves as both a filtering and pre-processing measure. To eliminate any potential ambiguity, we will revise the phrasing of the aforementioned statement in the manuscript.
>
> **Q- "More detailed analysis on why the proposed method fails to outperform DSSCCSBERT in BANKING and in KIR=25% setting"**
> **Response:** Please note that even in this specific case, our method still significantly outperforms DSSCCSBERT for the BANKING dataset on 2/3 metrics (i.e. ACC, ARI). We will include a more detailed analysis of this specific case in the final draft.
>
> "Reasons to reject
> 1. More evaluations for backbones other than RoBERTa are necessary such as BERT to prove the general improvement of the method for most pre-trained models.
> 2. More evaluations for backbones of more size are preferred."
>
> Although it is valuable to explore assessments using alternative backbones (varying models or sizes), our primary objective in this study is to ensure a direct comparison with the latest state-of-the-art models. Consequently, we meticulously adhere to the experimental parameters outlined in those baseline works.

---

### Meta-Review · Area_Chair_giM4 · 2023-10-04

**Recommendation:** 3

**Metareview:**

An unsupervised contrastive learning strategy for representation learning, even with interesting results, the paper lacks comprehensive experiments and a sound explanation of graph structure in the method section.

---

### Decision · Program_Chairs · 2023-10-07

**Decision:**

Accept-Findings

**Comment:**

An unsupervised contrastive learning strategy for representation learning, even with interesting results, the paper lacks comprehensive experiments and a sound explanation of graph structure in the method section.